# Genome-Wide Identification of Brassicaceae Hormone-Related Transcription Factors and Their Roles in Stress Adaptation and Plant Height Regulation in Allotetraploid Rapeseed

**DOI:** 10.3390/ijms23158762

**Published:** 2022-08-06

**Authors:** Shengjie Ma, Liwei Zheng, Xiaohan Liu, Kaiyan Zhang, Linlin Hu, Yingpeng Hua, Jinyong Huang

**Affiliations:** 1School of Agricultural Sciences, Zhengzhou University, Zhengzhou 450001, China; 2Zhengzhou Key Laboratory of Quality Improvement and Efficient Nutrient Use for Main Economic Crops, Zhengzhou 450001, China; 3School of Life Sciences, Zhengzhou University, Zhengzhou 450001, China

**Keywords:** Brassicaceae, phytohormone, transcription factors, abiotic stress, plant height

## Abstract

Phytohormone-related transcription factors (TFs) are involved in regulating stress responses and plant growth. However, systematic analysis of these *TFs* in Brassicaceae is limited, and their functions in stress adaptation and plant height (PH) regulation remain unclear. In this study, 2115 hormone-related *TFs* were identified in nine Brassicaceae species. Specific domains were found in several Brassicaceae hormone-related TFs, which may be associated with diverse functions. Syntenic analysis indicated that expansion of these genes was mainly caused by segmental duplication, with whole-genome duplication occurring in some species. Differential expression analysis and gene co-expression network analysis identified seven phytohormone-related *TFs* (*BnaWRKY7*, *21*, *32*, *38*, *52*, *BnaGL3-4*, and *BnaAREB2-5*) as possible key genes for cadmium (Cd) toxicity, salinity stress, and potassium (K) and nitrogen (N) deficiencies. Furthermore, *BnaWRKY42* and *BnaARR21* may play essential roles in plant height. Weighted gene co-expression network analysis (WGCNA) identified 15 phytohormone-related *TFs* and their potential target genes regulating stress adaptation and plant height. Among the above genes, *BnaWRKY56* and *BnaWRKY60* responded to four different stresses simultaneously, and *BnaWRKY42* was identified in two dwarf rapeseeds. In summary, several candidate genes for stress resistance (*BnaWRKY56* and *BnaWRKY60*) and plant height (*BnaWRKY42*) were identified. These findings should help elucidate the biological roles of Brassicaceae hormone-related *TFs*, and the identified candidate genes should provide a genetic resource for the potential development of stress-tolerant and dwarf oilseed plants.

## 1. Introduction

Plant hormones (phytohormones) are chemical messengers that coordinate plant growth and stress adaptation [1]. Phytohormones include auxin (IAA), cytokinin (CTK), abscisic acid (ABA), gibberellin (GA), ethylene (ET), brassinosteroid (BR), jasmonic acid (JA), salicylic acid (SA), and strigolactones (SL) [2]. Phytohormone-related transcription factors (TFs) [e.g., auxin response factors (ARFs) for IAA [3]; response regulators (RRs) for CTK [4]; ABA-insensitive (ABIs) and ABA-responsive element-binding (AREBs) for ABA [5]; gibberellin-insensitive (GAIs), rapid geriatric assessment (RGAs), and RGA-LIKE (RGLs) for GA [6]; ethylene-insensitive (EINs) and ethylene-insensitive-like (EILs) for ET [7]; brassinazole-resistant (BZRs) and BES1/BZR1 HOMOLOG (BEHs) for BR [8]; MYB21, MYB24, MYB75, MYC2, MYC3, MYC4, and GLABRA3 (GL3) for JA [9]; WRKY18, WRKY38, WRKY53, WRKY54, WRKY58, WRKY59, WRKY66, and WRKY70 for SA [10]; and TCP12 and TCP18 for SL [11] play essential roles in phytohormone signal transduction, plant growth, and stress adaptation.

ARFs regulate auxin response genes during plant growth and defense. For example, *arf2* mutants contain large seeds and are drought-tolerant [12], while overexpression of *TaARF4* in *Arabidopsis* causes shortened primary root length and plant height [13]. RRs for CTK can be categorized into types A and B according to their structure, expression pattern, and function [14]. Type-A *RR* genes, such as *ARR4* and *ARR5*, can be up-regulated by drought, salinity, and cold temperatures [15]. Overexpression of type-B *OsORR2* results in dwarfism in rice [16]. There are two types of ABA-related TFs (ABIs and AREBs) in *Arabidopsis*. Lateral root growth and photosynthesis are under the control of *ABI5* [17], while *AREB1* participates in the regulation of drought tolerance [18]. Five DELLA proteins, including GAI, RGA, RGLI, RGL2, and RGL3, mediate GA signal transduction [19], and play essential roles in promoting internode elongation [20]. In addition, EIN3 and EIL1 mediate ethylene responses, including stem elongation, maturation, senescence, and stress adaptation [21,22].

As a class of phytohormone, BR controls growth and stress tolerance. BZR1 and BES1 are critical components of the BR signaling pathway [8] and participate in drought resistance positively and negatively [23]. *BES1* and *BZR1* regulate plant architecture and their mutants are dwarfs [24]. JAs are well-recognized stress-related hormones and are involved in plant growth and development [25]. MYB21, MYB24, MYB75, GL3, and MYC2-4 are reported to influence JA signals, cell differentiation, and plant defense reactions [26]. SA affects seed germination, cell growth, stomatal closure, stress response, nitrogen fixation, senescence, and seed setting rate [27]. Six WRKY TFs (WRKY18, WRKY38, WRKY53-54, WRKY58-59, WRKY66 and WRKY70) mediate SA signals [10,28]. As a class of carotenoid-derived plant hormones, SLs affect branching, plant height (PH), drought adaptation, and phosphate starvation. Furthermore, TCP12 and TCP18 in the SL signaling pathway regulate plant architecture [29].

Abiotic stresses, including salinity, drought, nutrient deficiencies, and metal toxicity, severely affect agricultural crop productivity [30], and Brassicaceae is no exception. For instance, drought and high salinity can result in reduced biomass, seed germination, and shoot and root growth in *Brassica napus* and *Brassica oleracea* [31]. Plants have developed multiple mechanisms to respond to damage caused by stress, including transcription and regulation. For example, inducible expression of *ARR2* enhances drought and freezing tolerance in *Arabidopsis* [32]; *BnaWRKY47* contributes to the adaptation of rapeseed to B deficiency [33]; and BnaRGA proteins play important roles in *B. napus* adaptation to water-deficit stress [34].

Brassicaceae plants are important vegetable and oil crops [35]. Phytohormone-related *TFs* play crucial roles in plant growth and stress adaptation. However, the systemic study of *TFs* in Brassicaceae remains scarce. In this study, comprehensive analyses of *TFs* were performed in nine Brassicaceae species, including *Arabidopsis thaliana* (*A. thaliana*), *Brassica napus* (*B. napus*), *Brassica carinata* (*B. carinata*), *Brassica juncea* (*B. juncea*), *Brassica nigra* (*B. nigra*), *Brassica oleracea* (*B. oleracea*), *Brassica rapa* (*B. rapa*), *Capsella rubella* (*C. rubella*) and *Camelina sativa* (*C. sativa*). Chromosome locations, gene/protein structures, and phylogenetic and syntenic analyses were determined. Their potential involvement in regulating Cd, salt, K and N stress, and PH were investigated in allotetraploid rapeseed. The findings of this study should provide a valuable resource for Brassicaceae hormone-related *TFs*.

## 2. Results

### 2.1. Identification, Conserved Domain, and Gene Structure Analyses of Phytohormone-Related TF Genes in Brassicaceae

In total, 2115 phytohormone-related *TFs* were identified, including 82, 318, 314, 308, 223, 216, 203, 289, and 162 in *A. thaliana*, *B. napus*, *B. carinata*, *B. juncea*, *B. nigra*, *B. oleracea*, *B. rapa*, *C. sativa*, and *C. rubella*, respectively (Appendix A). Furthermore, we identified 363 IAA-, 473 CTK-, 123 ABA-, 70 GA-, 106 ET-, 143 BR-, 288 JA-, 454 SA-, and 96 SL-related *TFs* in the above species. These genes were named based on their chromosomal location (Appendix A).

In general, each type of Brassicaceae hormone-related TF contained conserved domains (Appendix A), with some exceptions. For example, nine additional domains (i.e., atrophin-1, HSF, COG5099) were found in IAA-related TFs, and 12 specific domains were found in CTK-related TFs, e.g., Med15 in BcaARR18 and A_thal in BjuARR40. Each class of phytohormone-related *TF* shared a similar gene structure. Of note, several *TFs* [e.g., *BcaTCP18-2* (12 kb, Appendix A) and *BcaARF10* (24 kb, Appendix A)] contained long introns.

### 2.2. Phylogenetic and Syntenic Analyses of Brassicaceae Hormone-Related TFs

Phylogenetic trees showed that most *B. napus*, *B. carinata*, *B. juncea*, *B. nigra*, *B. oleracea*, *B. rapa*, *C. sativa*, and *C. rubella* hormone-related TFs were highly homologous to the *Arabidopsis* TFs (Appendix A). In addition, the TFs were divided into distinct groups depending on phylogenetic analysis. For example, RGL3s were clustered in subgroup a, RGL2s in sub-group b, RGL1s in sub-group c, GAIs in sub-groups d and e, and RGAs in sub-group f (Appendix A).

To determine the expansion patterns of phytohormone-related *TFs*, duplication events in the nine Brassicaceae species were investigated (Appendix A). In total, 2030 duplicated gene pairs were found, including five, 359, 393, 386, 176, 165, 154, 357, and 40 pairs in *A. thaliana*, *B. napus*, *B. carinata*, *B. juncea*, *B. nigra*, *B. oleracea*, *B. rapa*, *C. sativa*, and *C. rubella*, respectively (Appendix A). To further understand the potential roles of unknown Brassicaceae hormone-related *TFs*, syntenic analysis between eight non-model Brassicaceae species and *Arabidopsis* was performed. In total, 120, 215, 223, 157, 145, 135, 206, and 90 gene pairs were identified in *A. thaliana-B. napus*, *A. thaliana-B. carinata*, *A. thaliana-B. juncea*, *A. thaliana-B. nigra A. thaliana-B. oleracea*, *A. thaliana-B. rapa*, *A. thaliana-C. sativa* and *A. thaliana-C. rubella* (Appendix A).

The ratio of non-synonymous to synonymous substitutions (Ka/Ks) was calculated to clarify divergence among orthologous gene pairs. The Ka/Ks ratios in most gene pairs were less than 1, except for *CsaWRKY48_CsaWRKY27*, *CsaARR14_CsaARR22*, *CsaMYB75-10_CsaMYB75-1*, *CsaWRKY48_CsaWRKY18*, *BcaMYB21/24-2_BcaGL3-6*, *BjuARR7_BjuARR69*, *BnaARR11_BnaARR52*, *BnaARR13_BnaARR56*, and *BnaTCP12-1_BnaTCP12-2* (Appendix A). The Ka/Ks ratios between gene pairs in Brassicaceae and *Arabidopsis* were all less than 1 (Appendix A).

### 2.3. Expression Profiles of B. napus Hormone-Related TFs in Response to Abiotic Stress

To identify the potential roles of rapeseed hormone-related *TFs* in adapting to abiotic stress, their responses to four stresses were investigated. FPKM (fragments per kilobase of transcript sequence per million mapped reads) was calculated to assess gene expression levels, and *p* < 0.05 and |log2(fold-change)| ≥ 1 were set as the criteria for identifying differentially expressed genes (DEGs). The DEGs were grouped into two clusters (1 and 2) according to their expression level. We then identified hub genes using Cytoscape (v3.8.2). Deleted genes with FPKM values < 1 and correlation value > 0.95 were used as the threshold for screening interactions between genes.

In shoots, 46 and 28 hormone-related *TFs* were increased and decreased, respectively, after Cd treatment (Figure 1a). Among these DEGs, *BnaWRKY7* was a core gene based on gene co-expression network analysis (GCNA) (Figure 1b). In roots, *BnaWRKY49*, *BnaWRKY38*, *BnaBZR11*, and *BnaBZR23* in cluster 1 were up-regulated by Cd, while *BnaARF33* and *BnaARR75* in cluster 2 were significantly reduced after Cd treatment (Figure 1c). *BnaWRKY38* may be a hub gene responding to Cd in roots (Figure 1d). Salinity severely inhibited rapeseed growth and yield [36]. In shoots, *BnaWRKY12* in cluster 1 was inhibited by salt, whereas *BnaWRKY49* in cluster 2 was induced (Figure 1e). Among them, *BnaAREB2-5* was identified as a hub gene in the gene co-expression network (Figure 1f). In roots, 86 phytohormone-related *TFs* were regulated by salt, with down-regulated *TFs* found in cluster 1 and up-regulated *TFs* found in cluster 2 (Figure 1g). Based on GCNA, *BnaWRKY52* played an essential role in salt adaptation (Figure 1h).

K is an essential nutrient for plant growth and development [37]. Here, K shortage altered the expression of 121 phytohormone-related *TFs* (Figure 2a,c). In shoots, genes in cluster 1 were induced; in contrast, genes in cluster 2 were inhibited (Figure 2a). Among them, *BnaWRKY21* was identified as a key gene (Figure 2b). Fewer genes were regulated by K stress in the roots (Figure 2c), and *BnaGL3-4* was identified as a likely core gene (Figure 2d). N is another important nutrient required for crop growth [38]. Here, two phytohormone-related *TFs* were under the control of N in shoots (Figure 2e). *BnaWRKY49* expression was markedly down-regulated under low N stress in roots, while seven other genes were induced (Figure 2f). Among the DEGs, *BnaWRKY32* was identified as a hub gene (Figure 2g).

### 2.4. Expression Profiles of Hormone-Related TFs in Rapeseed with Different PH and Stem Breaking Resistance (SBR)

Both PH and SBR are crucial agronomic traits [39]. Comparative transcriptome analysis between dwarf (df59, ed1, Ldt, and dwf) and wild-type (WT) rapeseeds was performed to define candidate genes. Compared with df59, nine hormone-related *TFs* were down-regulated in WT rapeseeds, while 18 were up-regulated (Figure 3a). As shown in Figure 3b, *BnaWRKY42* was identified as a hub gene among the above DEGs. In ed1, *BnaMYC2-2*, *BnaARR21*, and *BnaARR71* were down-regulated, while 12 other *TFs* (especially *BnaWRKY36* and *BnaWRKY59*) were up-regulated (Figure 3c). The key gene *BnaARR21* was appraised according to the gene co-expression network (Figure 3d). In Ldt, five rapeseed hormone-related *TFs* exhibited low expression (Appendix A). *BnaWRKY29*, *BnaAREB2-6*, and *BnaWRKY38* were clearly increased in dwf (Appendix A). Compared with rapeseeds with low SBR during flowering (FL), the expression levels of six phytohormone-related *TFs* were significantly altered with high SBR during flowering (FH) (Appendix A). Seven genes were down-regulated in rapeseeds with high SBR during silique development (SH) compared to those with low SBR (SL) (Appendix A).

### 2.5. Identification of Weighted Gene Co-Expression Network Analysis (WGCNA) Modules and Hub Genes Associated with Target Traits

WGCNA uses data from all genes to identify gene sets of interest, rather than genes only showing differential expression, and to analyze significant associations with phenotypes. WGCNA has two main advantages: i.e., loss of fewer genes and the ability to collate many genes into gene sets and identify their association with phenotypes without multiple hypothesis testing [40]. Therefore, we used WGCNA to analyze the RNA-sequencing data of Cd, salt, K, and N treatments and six dwarf mutants to identify hub genes and their target genes involved in stress adaptation and PH regulation.

Modules associated with Cd were identified using WGCNA, with the “green” module (r = −0.89 and *p* < 0.05) showing a high negative correlation with chlorophyll content (SPAD) and the “purple” module (r = 0.94 and *p* < 0.05) showing positive correlation with biomass (Figure 4a, Appendix A). A co-expression network was constructed to identify hub genes. *BnaWRKY60*, *BnaWRKY27*, *BnaWRKY56*, and *BnaARR39* were determined in response to Cd (Figure 4b,c). TFs regulate target genes by binding to specific *cis*-elements. Furthermore, WRKYs share the ability to bind to W-box *cis*-regulatory elements [41]. In the above two modules, heavy metal transport/detoxification superfamily proteins (BnaHMTSPs) were identified (Appendix A) [42]. In the “green” module, *BnaHMTSPs* (*BnaC07g28710D* and *BnaA07g36500D*) contained W-boxes and may be targets of BnaWRKY27, BnaWRKY56, and BnaWRKY60 (Figure 4d). *BnaARR39* likely regulates *BnaHMTSPs* (*BnaA06g39500D*, *BnaA07g36500D*, *BnaC07g10100D*, *BnaC07g28710D*, and *BnaC08g20630D*) through RR binding to *cis*-elements (Figure 4e). WGCNA was also applied to investigate the relationship between modules and salinity. The “salmon” (r = −0.83 and *p* < 0.05) and “blue” modules (r = −0.91 and *p* < 0.05) were negatively correlated with biomass and leaf area, respectively (Figure 5a). *BnaARR8* and *BnaARR14* were the two most important genes in the “salmon” module (Figure 5b and Appendix A), while *BnaBZR22*, *BnaAREB2-6*, *BnaAREB2-7*, and *BnaABI5/AREB1/2-2* were hub genes in the “blue” module (Figure 5c and Appendix A).

WGCNA was used to identify hub genes in response to K stress. As shown in Figure 6a, the “lightcyan” module (r = 0.92 and *p* < 0.05) was positively correlated with biomass, while the “turquoise” module (r = −0.92 and *p* < 0.05) was negatively correlated with SPAD. Gene interaction networks were constructed for these modules (Figure 6b,c and Appendix A). Three key genes (*BnaEIL1-2*, *BnaWRKY56*, and *BnaARR14*) were selected for their high connectivity (Figure 6b,c). Four K^+^ transport genes (*BnaKUP5*, two *BnaKUP6s* and *BnaPCP*) [43,44] were found in the “turquoise” module, and W-boxes and RR binding *cis*-elements were present in their promoters, suggesting they may be the targets of *BnaWRKY56* and *BnaARR14* (Figure 6d). Under N stress conditions, the “green” module (r = −0.83 and *p* < 0.05) was negatively correlated with SPAD (Figure 7a). *BnaBZR1* and *BnaBZR14* were identified as critical genes in the “green” module (Figure 7b and Appendix A).

The relationships between WGCNA modules and PH were also investigated. A total of 38 modules were obtained, with the “darkmagenta” module (r = −0.84 and *p* < 0.05) showing a negative correlation with PH (Figure 8a). We found several phytohormone-related *TFs* in the “darkmagenta” module (Appendix A). *BnaARF42* and *BnaARF26* with high connectivity were identified as hub *TFs* (Figure 8b). *ARFs* are reported to regulate PH [45]. Here, two ARF-binding *cis*-elements were found in *BnaARF10*, *BnaARF18*, and *BnaARF54*, indicating they were the targets of *BnaARF42* and *BnaARF26* (Figure 8c and Appendix A).

### 2.6. *Venn* Analyses of Phytohormone-Related TFs Mediating Stress Adaptation and PH Regulation

Based on differential expression analysis, DEG co-expression analysis, and WGCNA, two Venn diagrams were constructed to investigate the diverse roles of rapeseed hormone-related *TFs* in regulating Cd, salt, K, and N stress (Figure 9a), and PH (Figure 9b). Many *TFs* responded to the multiple stresses simultaneously. In total, 21 *TFs* (i.e., *BnaARF2*, *BnaARR18*, *BnaWRKY1*, *BnaWRKY2*, *BnaWRKY10*, *BnaWRKY12*, *BnaWRKY13*, *BnaWRKY15*, *BnaWRKY25*, *BnaWRKY29*, *BnaWRKY33*, *BnaWRKY34*, *BnaWRKY42*, *BnaWRKY44*, *BnaWRKY48*, *BnaWRKY53*, *BnaWRKY55*, *BnaWRKY56*, *BnaWRKY60*, *BnaWRKY62*, and *BnaWRKY63*) were affected by the four conditions (Appendix A). *BnaWRKY13* was identified as a DEG in FH/FL, ed1/WT, and SH/SL (Appendix A). *BnaWRKY38* likely regulated PH in df59 and dwf. *BnaWRKY44*, *BnaARR47*, and *BnaWRKY15* were DEGs in WT/df59 and Ldt/Lwt. *BnaWRKY42*, *BnaWRKY59*, *BnaWRKY33*, and *BnaARR71* may be involved in the dwarfing of df59 and ed1.

## 3. Discussion

### 3.1. Comparison of Plant Hormone-Related TFs among Brassicaceae

Genes within a family usually exhibit obvious variations during their evolutionary history, which contribute to gene family division, expansion, and functional divergence [46]. Brassicaceae separated from *Arabidopsis* approximately 43.2 million years ago and underwent genome triplication, with *B. napus* (2*n* = 38), *B. carinata* (2*n* = 34), and *B. juncea* (2*n* = 36) arising from the genome hybridization of *B. nigra* (2*n* = 16), *B. oleracea* (2*n* = 18), and *B. rapa* (2*n* = 20) [47]. New evidence suggests a possible link between polyploidy and enhanced stress tolerance [48]. Notably, amphidiploid species *B. juncea*, *B. napus*, and *B. carinata* are more tolerant to stress than diploids such as *B. oleracea*, *B. nigra*, and *B. rapa* [49]. There are 82 phytohormone-related *TFs* in *Arabidopsis* [50,51,52,53,54,55,56,57]. In our study, 318, 314, 308, 223, 216, 203, 289, and 162 hormone-related *TFs* were identified in *B. napus*, *B. carinata*, *B. juncea*, *B. nigra*, *B. oleracea*, *B. rapa*, *C. sativa* and *C. rubella* (non-model Brassicaceae species), respectively (Appendix A), nearly 3.8, 3.8, 3.7, 2.7, 2.6, 2.4, 3.5, and 1.9 times higher than that in *Arabidopsis*. Furthermore, more segmental hormone-related *TFs* were found in the non-model Brassicaceae species than in *Arabidopsis* (Appendix A); *B. napus*, *B. carinata*, *B. juncea*, and *C. sativa* have undergone whole-genome duplication [58,59,60,61,62]. Therefore, the expansion and evolution of *TFs* in the above species may have been induced by gene and genome-wide duplications. In addition, the number of SA- and SL-related *TFs* in the above eight species was more than 5.5 times higher than that in *Arabidopsis*, suggesting pivotal roles of SA and SL in the stress tolerance of these species.

Gene functions are always associated with conserved domains [63]. Several distinct domains, including C2, HSF, and RALF, were identified in some Brassicaceae hormone-related TFs (Appendix A). Notably, C2, which is associated with calcium-binding [64], was found in BolGL3-1; HSF, which participates in the regulation of biotic and abiotic stress [65], was found in BjuARF4; and RALF, which is involved in biotic and abiotic stress responses [66], was found in BniBZR1. These results indicate that most Brassicaceae hormone-related TFs shared conserved functions, while several genes may vary among Brassicaceae species.

A total of 1291 gene pairs were identified between *A. thaliana* and non-model Brassicaceae species (Appendix A), and the functions of Brassicaceae hormone-related *TFs* were predicted based on homologous *Arabidopsis TFs*. For instance, *AtARF6* and *AtBZR1* regulate hypocotyl and stem elongation in *Arabidopsis* [67] and homologous genes (*BcaARF20*, *CruARF4*, and *BolBZR12*) may share similar roles; *CsaARR63* and *BnaARR49* may control defense response in Brassicaceae based on the homologous gene *AtARR6* [68].

### 3.2. Putative Functions of B. napus Hormone-Related TFs in Regulating Stress Adaptation and PH

We found that most *BnaWRKY* DEGs were up-regulated, while *BnaARRs* were downregulated in both shoots and roots under Cd and K treatment (Figure 1a,c and Figure 2a,c). We speculated that Cd poisoning and K starvation may induce SA expression and inhibit CTK expression in rapeseed, thereby enhancing plant Cd resistance and K import and translocation. We also found that salt stress and N deficiency led to a significant decrease in the expression of most *BnaWRKYs* in the shoots and roots (Figure 1e,f and Figure 2e,f). Therefore, we concluded that salinity and limited N supply may restrict transcription of SA, thereby enhancing plant salt resistance and N uptake. Taken together, our results showed that most phytohormone *TFs* were responsive to diverse abiotic stresses, implying essential roles in the resistance or adaptation of rapeseed plants to stress.

Previous studies have shown that phytohormone-related *TFs* are essential for stress adaptation [69]. Various *TFs*, such as *AtWRKY13*, *SbWRKY50*, *AtARR22*, *SlBZR1*, and *AtAREB2*, are associated with plant survival under Cd and salt tolerance and K and N starvation [70,71,72]. However, whether they respond to Cd, salt, K, and N stress and regulate PH in rapeseed remains unknown. Here, candidate phytohormone-related *TFs* were identified in rapeseed through DEG co-expression analysis and WGCNA. *BnaWRKY7*, *27*, *38*, *56*, *60*, and *BnaARR39* were identified as key Cd-related genes (Figure 1b,d and Figure 4). *BnaWRKY52*, *BnaBZR22*, *BnaARR8*, *14*, *BnaAREB2-5*, *-6*, *-7*, and *BnaABI5/AREB1/2-2* were identified as salt stress candidate genes (Figure 1f,h and Figure 5). *BnaGL3-4*, *BnaARR14*, *BnaWRKY21*, *56*, and *BnaEIL1-2* were identified as K deficiency candidate genes (Figure 2b,d and Figure 6). *BnaWRKY32*, *BnaBZR1*, and *BnaBZR14* were identified as responsive to N stress (Figure 2g and Figure 7). *AtWRKY53* and *AtARR10* are reported to negatively regulate plant responses to drought and salt stress [73,74]. Based on orthologous relationship analysis, we found that the orthologous gene of *AtWRKY53* in *B. napus* was *BnaWRKY7* and the orthologous gene of *AtARR10* was *BnaARR8*.

Various genes and gene families involved in PH regulation have been characterized in plants. Different from our study, however, previous research has primarily focused on specific *TFs*. In *A. thaliana*, *WRKY46*, *WRKY54*, *WRKY70*, *ARF6*, and *BZR1* regulate cell elongation and PH [75,76]. To further analyze the regulation of phytohormone *TFs* on PH, we performed DEG co-expression analysis and WGCNA. Results revealed that PH was mediated by *BnaWRKY42*, *BnaARR21*, *BnaARF26*, and *BnaARF42* (Figure 3b,d and Figure 7). Therefore, these genes are likely important for oilseed architecture and stress adaptation.

K- and Cd-related genes (*BnaWRKY56* and *BnaWRKY60*) identified through WGCNA (Figure 4 and Figure 6) responded to all four stress conditions. Furthermore, most key genes revealed through DEG co-expression analysis and WGCNA may play core roles in regulating oilseed resistance to three stresses (Figure 9a). *BnaWRKY38*, *BnaWRKY42*, and *BnaWRKY59*, which may regulate PH, were identified in two dwarf mutants (Figure 9b). In addition, *BnaWRKY38* was identified as a key gene in response to Cd stress, and *BnaWRKY42* was differentially expressed under all four stress conditions. This suggests that the phytohormone *TFs* can simultaneously regulate PH and stress adaptation. Therefore, future studies should focus on the potential functions of these key genes.

## 4. Methods and Materials 

### 4.1. Identification and Chromosome Locations of Hormone-Related TFs in Brassicaceae

The protein sequences of *Arabidopsis* hormone-related TFs were used as queries to BLAST the *B. napus*, *B. oleracea*, *B. rapa*, *B. nigra*, *B. juncea*, *B. carinata*, *C. rubella*, and *C. sativa* genomes. We retrieved the phytohormone TF gene sequences using the following databases: Arabidopsis Information Resource (TAIR10, https://www.arabidopsis.org/, accessed on 15 May 2022) for *Arabidopsis*, Genoscope (http://www.genoscope.cns.fr/brassicanapus/, accessed on 15 May 2022) for *B. napus*, Bol base v1.0 (http://119.97.203.210/bolbase/index.html, accessed on 15 May 2022) for *B. oleracea*, Brassica Database (BRAD) v1.1 (http://brassicadb.org/brad/, accessed on 15 May 2022) for *B. carinata*, *B. rapa*, *B. juncea*, and *B. nigra*, National Center for Biotechnology Information (NCBI, www.ncbi.nlm.nihgov, accessed on 15 May 2022), EnsemblPlants (http://plants.ensembl.org/indexhtml, accessed on 15 May 2022), and Phytozome v10 (http://phytozome.jgi.doegov/pz/portal.html, accessed on 15 May 2022) for *C. sativa* and *C. rubella*. The characteristic domains were confirmed using the Batch Web CD-Search Tool (https://www.ncbi.nlm.nih.gov/Structure/bwrpsb/bwrpsb.cgi, accessed on 15 May 2022) and visualized using the “Gene Structure View (Advanced)” tool in TBtools (v1.09876) to confirm their highly conserved segments. The “Gene Location Visualize from GTF/GFF” tool in TBtools (v1.09876) was used to visualize chromosomal locations, named according to their chromosome order [77].

### 4.2. Phylogeny, Gene Structure, and Syntenic Analyses of Brassicaceae Hormone-Related TFs

After aligning the full-length protein sequences using ClustalW with default parameters, MEGA X (v10.1.6, University of Pennsylvania, Philadelphia, PA, USA) was used to construct the phylogenetic tree with the maximum-likelihood method. Using generic feature format version 3 files of *TFs*, gene structure analyses were completed using ‘Visualize Gene Structure (Advanced)’ in TBtools (v1.09876). We performed syntenic analysis, with results visualized using the “One Step MCScanX” and “Advanced Circos” functions in TBtools (v1.09876) [77]. Genes were determined as duplicates in each genome based on the following [78]: (1) Aligned gene sequences were more than 70% identical, and the length of matching sequences was at least 70% of the longer gene; (2) Duplicates on different chromosomes were characterized as segmental duplications. The “Simple Ka/Ks Calculator (NG)” function in TBtools (v1.09876) was used to calculate Ka, Ks, and Ka/Ks values.

### 4.3. Transcriptome Analysis, GCNA, and WGCNA of Brassicaceae Hormone-Related TFs

The transcriptome data can be found in published papers [79,80,81,82,83,84,85,86], and all the data that are required to reproduce these findings can be shared by contacting the corresponding author. Using fastp software (v0.20.1), we evaluated the overall sequencing quality of raw reads and removed low-quality reads. Hisat2 (v2.1.0) and SAMtools (v1.6) were used to align the high-quality reads to the *B. napus* reference genome sequence (http://cbi.hzau.edu.cn/cgi-bin/rape/download_ext, accessed on 15 May 2022). The expression levels of high-confidence genes in each sample were calculated with Stringtie (v1.3.3b). DEGs were defined using the R package “edgeR”, with *p* < 0.05, false-discovery rate (FDR) < 0.05, and |log2(fold change)| ≥ 1. GCNA was completed with the cor. test function in R (v4.1) and visualized with Cytoscape (v3.8.2, https://cytoscape.org/download.html, accessed on 13 April 2022) based on our previously described protocol [87]. According to our earlier study [87], the R WGCNA package (v1.51) was used to perform WGCNA with high-quality genes. By calculating the module eigengene value, significant module-trait relationships with PH, SPAD, biomass, and leaf area were identified. Cytoscape (v3.8.2, https://cytoscape.org/download.html, accessed on 13 April 2022) was used to visualize gene co-expression networks.

## 5. Conclusions

In this study, 2115 phytohormone-related *TFs* were systematically identified in nine Brassicaceae species. Their chromosome locations, gene/protein structures, and phylogenetic and syntenic relationships were characterized. Genes responding to Cd, salt, K, and N adaptation in *B. napus* were investigated through differential expression analysis and DEG co-expression network analysis. In addition, WGCNA indicated that 15 and two phytohormone-related *TFs* and their potential target genes responded to stress and PH regulation, respectively. Taken together, *BnaWRKY56* and *BnaWRKY60* were identified as potential hub genes of rapeseed resistance to stress. *BnaWRKY42* may play an essential role in regulating PH. Our results showed that SA-related *BnaWRKY TF**s* may be crucial for regulating PH and stress adaptation in rapeseed. The above-mentioned candidate genes should be validated in future studies.

## Figures and Tables

**Figure 1 ijms-23-08762-f001:**
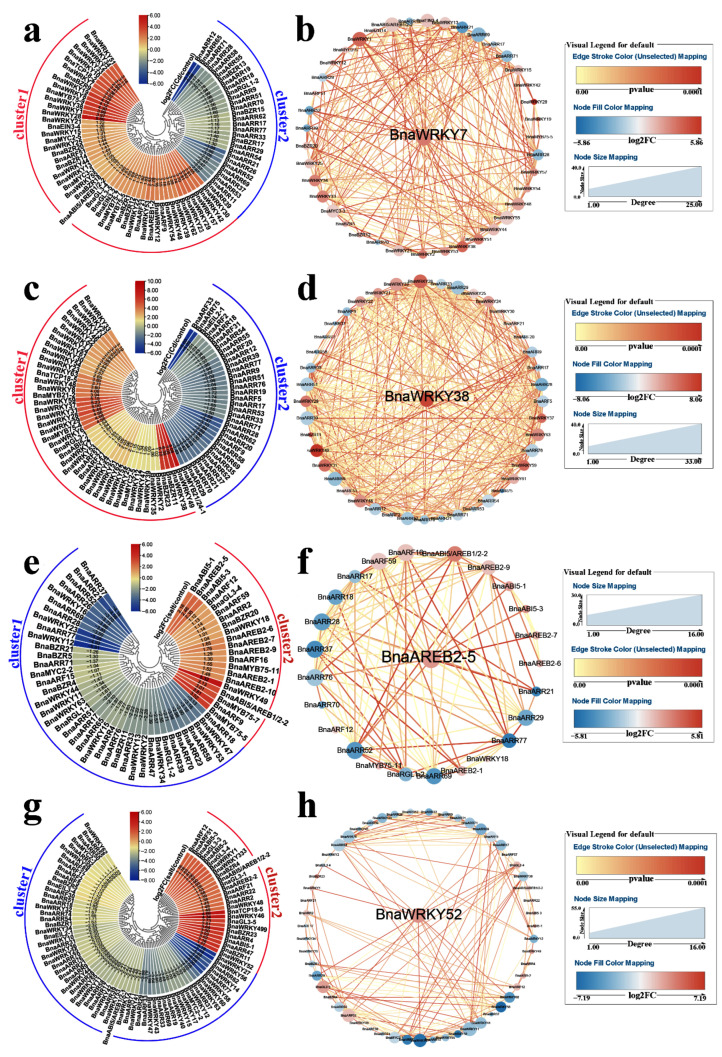
Expression profiles of *B. napus* hormone-related *TFs* in response to Cd and salt toxicity. Cycle nodes represent genes and the size of the node represents the power of the inter-relationship among nodes by degree value; Color of the node represents log2FC value; Red indicates up-regulated genes and blue indicates down-regulated genes; Edges between nodes represent *p*-value. (**a**) Expression analysis of *B. napus* hormone-related *TFs* in response to Cd toxicity in shoots. (**b**) Co-expression network analysis of differentially expressed *B. napus* hormone-related *TFs* in response to Cd toxicity in shoots. (**c**) Expression analysis of *B. napus* hormone-related *TFs* in response to Cd toxicity in roots. (**d**) Co-expression network analysis of differentially expressed *B. napus* hormone-related *TFs* in response to Cd toxicity in roots. (**e**) Expression analysis of *B. napus* hormone-related *TFs* in response to salt toxicity in shoots. (**f**) Co-expression network analysis of *B. napus* hormone-related *TFs* in response to salt toxicity in shoots. (**g**) Expression analysis of *B. napus* hormone-related *TFs* in roots in response to salt toxicity. (**h**) Co-expression network analysis of *B. napus* hormone-related *TFs* in response to salt toxicity in roots.

**Figure 2 ijms-23-08762-f002:**
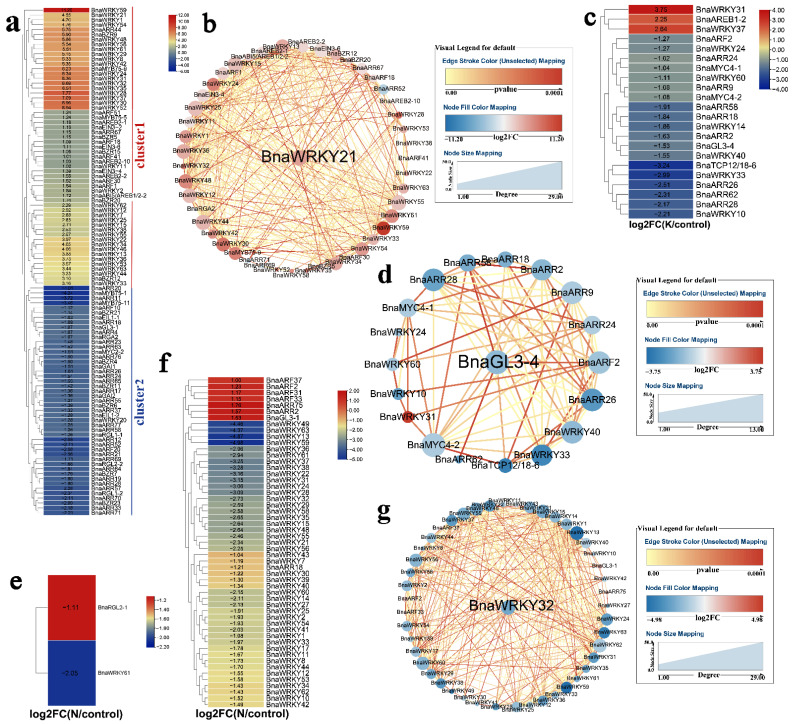
Expression profiles of *B. napus* hormone-related *TFs* in response to K and N starvation. Cycle nodes represent genes and the size of the node represents the power of the inter-relationship among nodes by degree value; Color of the node represents log2FC value; Red indicates up-regulated genes and blue indicates down-regulated genes; Edges between nodes represent *p-*value. (**a**) Expression analysis of *B. napus* hormone-related *TFs* in response to K starvation in shoots. (**b**) Co-expression network analysis of differentially expressed *B. napus* hormone-related *TFs* in response to K starvation in shoots. (**c**) Expression analysis of *B. napus* hormone-related *TFs* in response to K starvation in roots. (**d**) Co-expression network analysis of differentially expressed *B. napus* hormone-related *TFs* under in response to K starvation in roots. (**e**) Expression analysis of *B. napus* hormone-related *TFs* in response to N starvation in shoots. (**f**) Expression analysis of *B. napus* hormone-related *TFs* in response to N starvation in roots. (**g**) Co-expression network analysis of differentially expressed *B. napus* hormone-related *TFs* in response to N starvation in shoots and roots.

**Figure 3 ijms-23-08762-f003:**
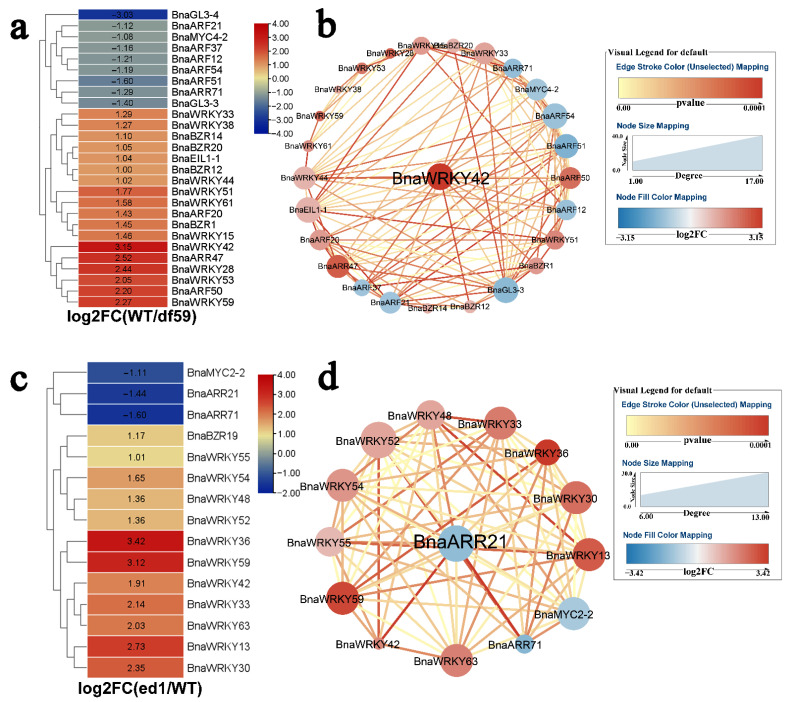
Expression profiles of *B. napus* hormone-related *TFs* in response to PH. Cycle nodes represent genes and the size of the node represents the power of the inter-relationship among nodes by degree value; Color of the node represents log2FC value; Red indicates up-regulated genes and blue indicates down-regulated genes; Edges between nodes represent *p*-value. (**a**) Expression analysis of *B. napus* hormone-related *TFs* in a conventional rapeseed cultivar (Ningyou 18) and a dwarf mutant (df59). (**b**) Co-expression network analysis of differentially expressed *B. napus* hormone-related *TFs* in WT and df59. (**c**) Expression analysis of hormone-related *TFs* in an extreme dwarf mutant of rapeseed (ed1) and WT. (**d**) Co-expression network analysis of differentially expressed *B. napus* hormone-related *TFs* in ed1 and WT.

**Figure 4 ijms-23-08762-f004:**
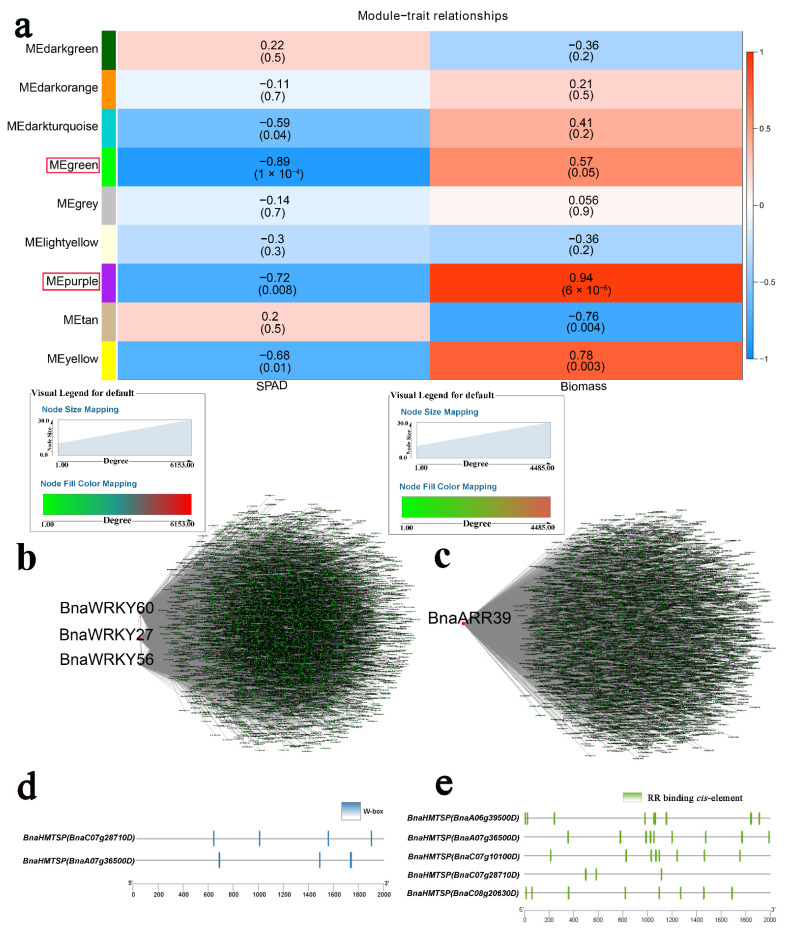
WGCNA of rapeseed genes in response to Cd stress. (**a**) Module-trait correlation showing the significance of module eigengene correlation with traits (SPAD and biomass). Left panel shows modules. (**b**) Cytoscape representation of the relationship of hormone-related *TFs* in “green” module. Key genes are represented by large red circles. (**c**) Cytoscape representation of the relationship of hormone-related *TFs* in “purple” module. Key genes are represented by large red circles. (**d**) W-box *cis*-element in promoter of *BnaHMTSPs*. (**e**) RR binding *cis*-element genes in promoter of *BnaHMTSPs*.

**Figure 5 ijms-23-08762-f005:**
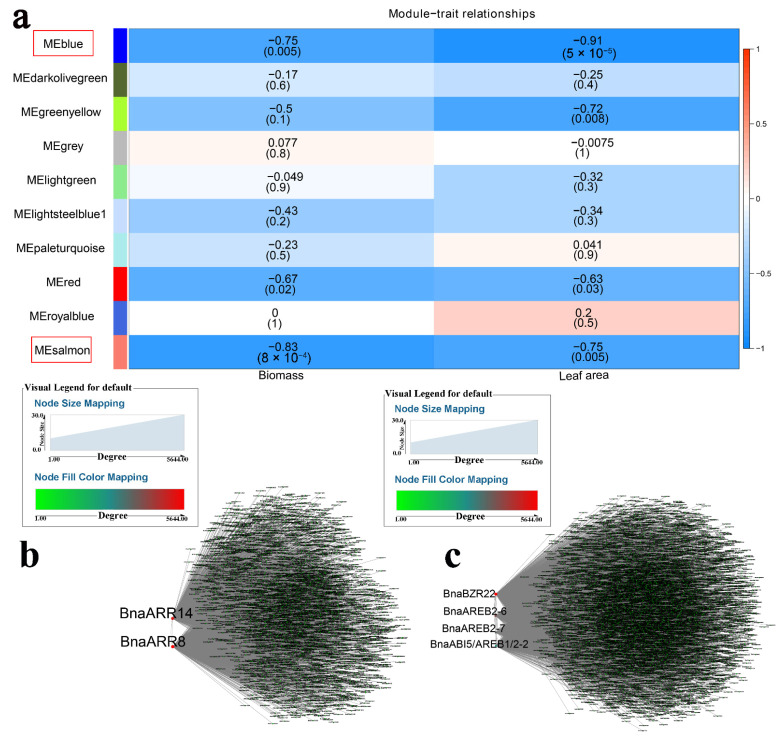
WGCNA of rapeseed genes in response to salt. (**a**) Module-trait correlation showing the significance of module eigengene correlation with traits (biomass and leaf area). Left panel shows modules. (**b**) Cytoscape representation of the relationship of hormone-related *TFs* in “salmon” module. Key genes are represented by large red circles. (**c**) Cytoscape representation of the relationship of hormone-related *TFs* in “blue” module. Key genes are represented by large red circles.

**Figure 6 ijms-23-08762-f006:**
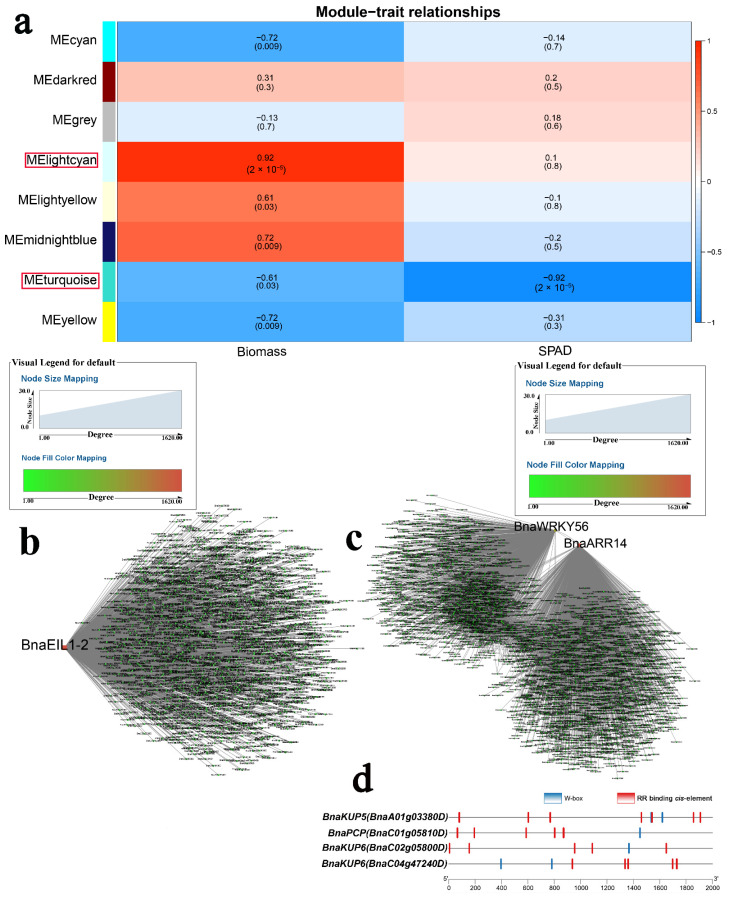
WGCNA of rapeseed genes in response to K starvation. (**a**) Module-trait correlation showing the significance of module eigengene correlation with traits (biomass and SPAD). Left panel shows modules. (**b**) Cytoscape representation of the relationship of hormone-related *TFs* in “lightcyan” module. Key genes are represented by large red circles. (**c**) Cytoscape representation of the relationship of hormone-related *TFs* in “turquoise” module. Key genes are represented by large red circles. (**d**) W-box and RR binding *cis*-element in promoters of *BnaKUP5*, *BnaPCP*, and *BnaKUP6*.

**Figure 7 ijms-23-08762-f007:**
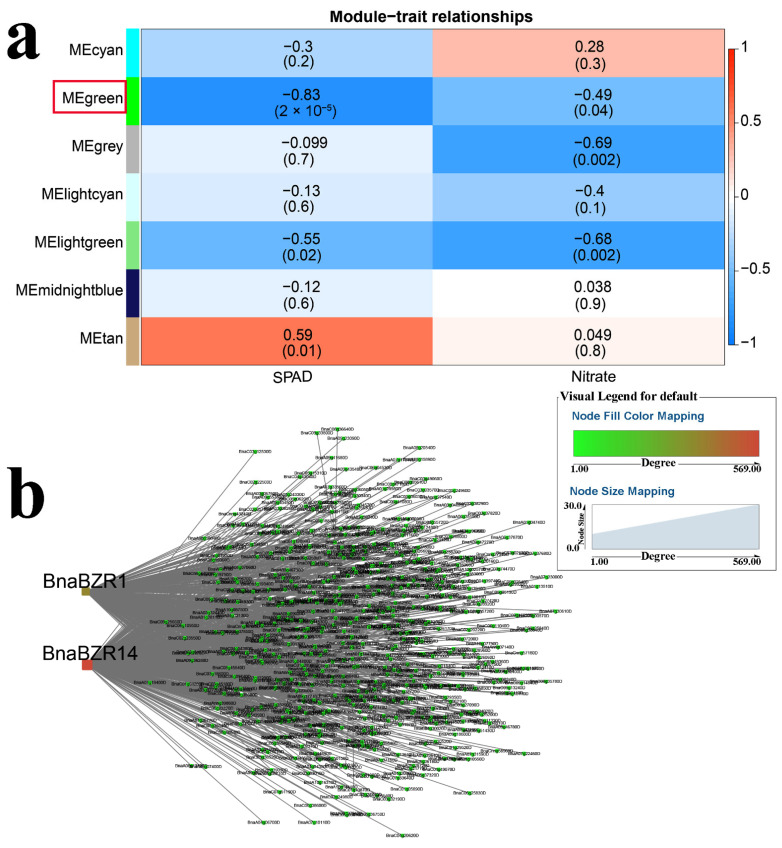
WGCNA of rapeseed genes in response to N deficiency. (**a**) Module-trait correlation showing the significance of module eigengene correlation with traits (SPAD). Left panel shows modules. (**b**) Cytoscape representation of the relationship of hormone-related *TFs* in “green” module. Key genes are represented by large red circles.

**Figure 8 ijms-23-08762-f008:**
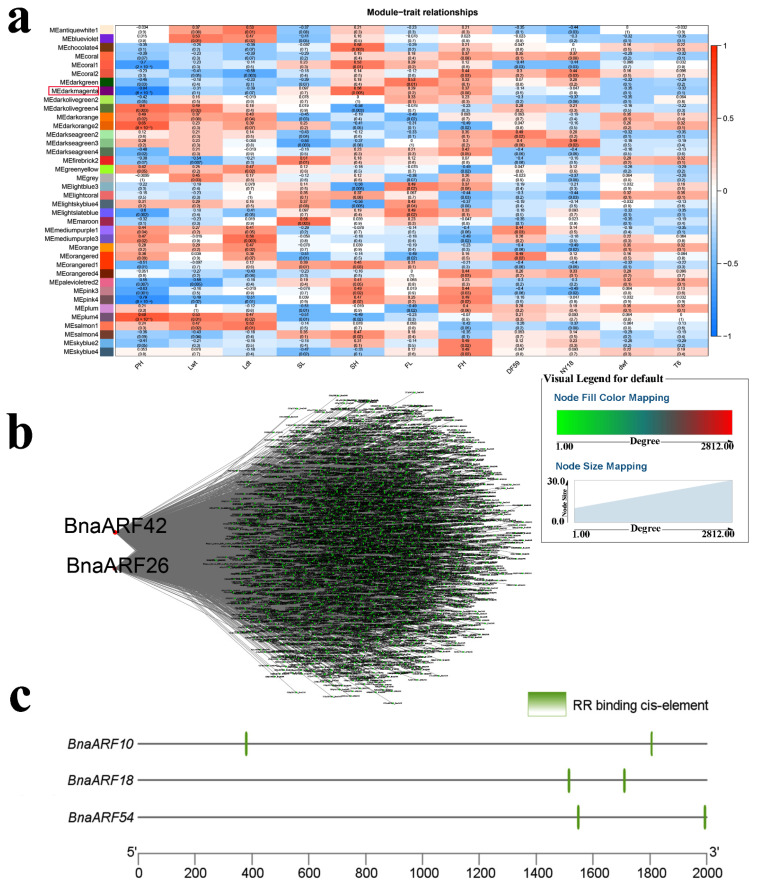
WGCNA of rapeseed genes identified in dwarf mutants. (**a**) Module-trait correlation showing the significance of module eigengene correlation with traits (PH). Left panel shows modules. (**b**) Cytoscape representation of the relationship of hormone-related *TFs* in “darkmagenta” module. Key genes are represented by large red circles. (**c**) RR-binding *cis*-element in the promoter of *BnaARFs*.

**Figure 9 ijms-23-08762-f009:**
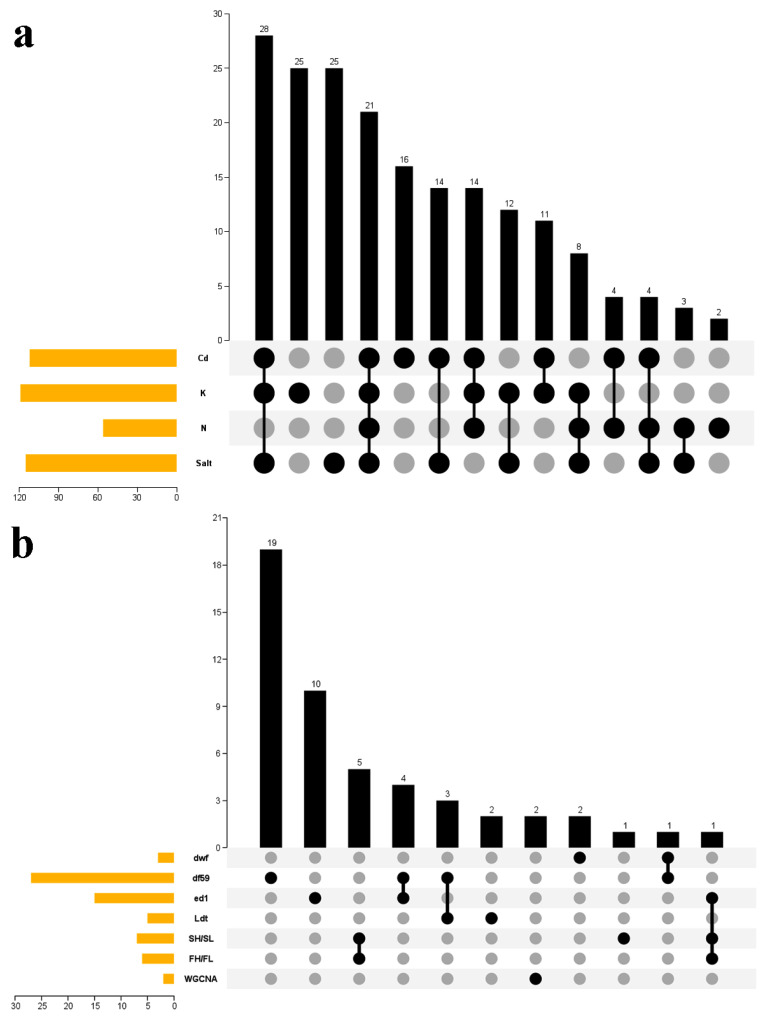
Venn diagram indicating various functions of *B. napus* hormone-related *TFs* in stress adaptation and PH regulation. (**a**) Number of rapeseed hormone-related *TFs* in response to Cd, K, N, and salt stress. (**b**) Number of rapeseed hormone-related *TFs* identified in rapeseed dwarf mutants.

## Data Availability

The datasets used in this study can be found in published papers. The datasets used and/or analyzed in the current study are available from the corresponding author upon reasonable request.

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
