# Peer review of "Genome-Wide Identification of Brassicaceae Hormone-Related Transcription Factors and Their Roles in Stress Adaptation and Plant Height Regulation in Allotetraploid Rapeseed"

_ijms, 2022, doi:10.3390/ijms23158762_

Round 1

Reviewer 1 Report

The paper focuses on identifying hormone-related transcription factor genes that regulate stress responses from different Brassicaceae species.

The introduction contains sufficient information about the plant's hormones and the genes which regulate them. It will be essential to add a paragraph related to the responses of Brassicaceae species to different stress.

The results are very detailed and completed.

The discussions need to be much more detailed. The discussions about figures from the results are really necessary.

The methods are appropriate.

The conclusions are trivial and have to be improved. It will be better to add some correlations between different genes.

Reviewer 2 Report

In this study, authors conducted intensive bioinformatic analyses to identify hormone related transcription factors in the family Brassicaceae. In general, the purpose of the study is fine for publication. However, authors did not explain about the figures in detail. Figure legends should be detailly written. The quality of figures was very poor and too small. Please provide high quality images. All characters in the images should be readable. Authors did not describe about methods in detail. Discussion was too short. My comments are as follows.  

L24-28 Please divide the sentence into several sentences. Too long. Please spell out Cd, K, and N. There are too many “And” in this sentence. I found many “And” in the manuscript. Please delete unnecessary words.

L30 in response to four different stresses

L31 delete “And”.

L55 RRs for CTK

L59 were dwarf in “what plant species”.

L86 PH -> pH

Figure 1 should be explained in detail. For example, what are cluster 1 and cluster 2?

The size of figures was too small. Please magnify them. Please explain about expression level based on the colors. In co-expression network, only representative genes were shown. Please explain this in the figure legend. The additional panels in Figure 1b, 1d, 1f, 1h should be described.

Some characters in the figure 2 were too small. Please increase the size of images.

In Figure 3, the images in the right-upper for Figure 3b and 3d were too small.

L227 “genes with a large and small degree are indicated with large red and small green circles.” I don’t understand it. Please explain this.

In Figure 4, I cannot read “visual legend XYZ” in the b and c.

L204-207 Please explain about WGCNA and its results in detail. Especially, please explain about modules. Nobody understands it.

Please prepare high quality figures for Figure 4, 5, 6, and 7. Please provide detailed explanation for all figures.

Discussion was too short. Please discuss about main findings and novelty of this study as compared to other previous studies.

L353-356 Please write scientific names in italics.

In methods

Please describe the name, website, and reference for the programs used for data analyses. For example, with “Simple Ka/Ks Calculator (NG)”, I could not find any related software.

In addition, data analyses should be explained in detail. Authors just listed the name of softwares. Please provide full descriptions about methods with respective parameters.

The data sources were not well described. “The transcriptome data can be found in published papers [72-81]” is not enough. Please provide list of all SRA data as a table used in this study.

Reviewer 3 Report

1- High resolution images for most of the figures are badly required since the contents of the figures are not readable so I recommend that unnecessary data should go to supplementary figures so the main figures are less crowded

2- Language, grammar, and punctuation marks require further check

3- I would recommend citing these references since they are tightly linked to impact of polyploidy and response to stress

A- Yahya, G., Menges, P., Ngandiri, D.A., Schulz, D., Wallek, A., Kulak, N., et al. (2021) Scaling of cellular proteome with ploidy.

B- Kamal, K. Y., Khodaeiaminjan, M., Yahya, G., El-Tantawy, A. A., Abdel El-Moneim, D., El-Esawi, M. A., Abd-Elaziz, M., & Nassrallah, A. A. (2021). Modulation of cell cycle progression and chromatin dynamic as tolerance mechanisms to salinity and drought stress in maize. Physiologia plantarum, 172(2), 684–695. https://doi.org/10.1111/ppl.13260

Round 2

Reviewer 2 Report

Authors properly revised their manuscript according to reviewer's comments except the following comment.

Please provide accession numbers used for this study. If some data are not available, then indicate those data can be accessible by the corresponding author. 

19. The data sources were not well described. “The transcriptome data can be found in published papers [72-81]” is not enough. Please provide list of all SRA data as a table used in this study.

Thank you for your suggestion. It is of great value. However, as the relevant research is still in progress, the author has not uploaded the transcriptome data. All the data that are required to reproduce these findings can be shared by contacting the corresponding author.

Author Response

  1. Please provide accession numbers used for this study. If some data are not available, then indicate those data can be accessible by the corresponding author.

According to your suggestion, we have indicated that all the data can be shared by contacting the corresponding author.

Reviewer 3 Report

The authors fulfilled most of the comments and greatly improved the manuscript. I am pleased to accept the manuscript in its current form for publication

Author Response

Thank you for the time and effort in handling the manuscript.